# Trait-based community assembly and succession of the infant gut microbiome

John Guittar [1], Ashley Shade [2,3] & Elena Litchman[1,4]

The human gut microbiome develops over early childhood and aids in food digestion and immunomodulation, but the mechanisms driving its development remain elusive. Here we use data curated from literature and online repositories to examine trait-based patterns of gut microbiome succession in 56 infants over their first three years of life. We also develop a new phylogeny-based approach of inferring trait values that can extend readily to other microbial systems and questions. Trait-based patterns suggest that infant gut succession begins with a functionally variable cohort of taxa, adept at proliferating rapidly within hosts, which gradually matures into a more functionally uniform cohort of taxa adapted to thrive in the anoxic gut and disperse between anoxic patches as oxygen-tolerant spores. Trait-based composition stabilizes after the first year, while taxonomic turnover continues unabated, suggesting functional redundancy in the traits examined. Trait-based approaches powerfully complement taxonomy-based approaches to understanding the mechanisms of microbial community assembly and succession.

---

[1] Kellogg Biological Station, Michigan State University, 3700 E Gull Lake Dr., Hickory Corners, MI 49060, USA. [2] Department of Microbiology and Molecular Genetics, Department of Plant, Soil and Microbial Sciences, Michigan State University, East Lansing, MI 48840, USA. [3] Program in Ecology, Evolutionary Biology and Behavior, Michigan State University, East Lansing, MI 48840, USA. [4] Department of Integrative Biology, Michigan State University, East Lansing, MI 48824, USA. Correspondence and requests for materials should be addressed to J.G. (email: guittarj@msu.edu)

Classical ecological theory posits that successional patterns arise from the combined influence of dispersal, species interactions, and the environment[1,2], and this general framework extends readily to gut communities[3]. Before a microbe can inhabit the colon, the most distal and speciose part of the gastrointestinal tract, it must first be swallowed by the host and survive the acidic conditions of the stomach and small intestine (i.e., it must disperse). A species will persist in the colon only if it can acquire enough resources to reproduce (i.e., it must be competitive) or arrive there in high enough numbers to sustain a population[4]. Microbial colonists may then alter the environment, e.g., by depleting intestinal oxygen[5] or providing opportunities for cross-feeding[6], favoring taxa with different phenotypes as succession proceeds.

Yet successional patterns in the gut may differ from classical successional expectations due to the active influence of the host and the host mother[7,8]. Early colonists are passed directly from the mother during or even before birth[9], and therefore may lack characteristics that would otherwise facilitate early arrival, e.g., via active dispersal, and instead have characteristics selected for in the mother's gut or vaginal environment. Following birth, mothers supply bacterial growth factors in breast milk and continue to introduce new taxa through physical contact[10]. Meanwhile, the maturing infant is beginning to suppress undesirable taxa through immune response[11], and actively cultivate commensal taxa by providing nitrogen-rich mucus and favorable habitat in the outer mucus layer of the large intestine[12]. Gut community composition is also affected by the introduction of solid food[13], in particular with the introduction of insoluble fiber[14].

One approach to determining the relative influence of different mechanisms of community assembly is to examine patterns in trait-based community composition[15]. A trait is commonly defined as a measurable organismal characteristic directly or indirectly linked to fitness or performance[16]. As such, observable shifts in the trait-based composition of a community imply shifts in local environmental conditions favoring different species and/or dispersal limitation (i.e., when a taxon does not colonize a site because it does not arrive). Despite the success and proliferation of trait-based approaches to study community assembly in plant[17,18], animal[19,20], and phytoplankton systems[21], they have only rarely been used for bacterial and archaeal systems[22,23]. This is due partly to the challenges of identifying ecologically relevant traits for a functionally diverse cohort of taxa, and partly to a dearth of curated trait data. But thanks to recent advances in high-throughput molecular techniques, renewed efforts to directly collect phenotypic data[24], and the aggregation of data from disparate sources[25,26], trait-based approaches to microbial community dynamics are becoming more feasible, especially for well-studied systems like the human gut.

Here, we examine trait-based successional patterns in a cohort of 56 infants from Finland and Estonia for which longitudinal microbiome survey data were publicly available[27,28]. We develop a unique approach to inferring microbial trait data, which entails (1) building a phylogeny that contains the taxa from infant gut samples and 13,900 other taxa with formally described type specimens and Latin binomials[29], (2) using the Latin binomials to map trait data curated from literature and online repositories onto the tips of the phylogeny, and (3) inferring unknown trait values using hidden state prediction. We then compare taxonomic and trait-based community turnover in time (i.e., over infant development) and space (i.e., across infants) to gain insight into the mechanisms driving successional patterns. We show significant trends in predicted traits over the first year of infant development, during which time oxygen-tolerant taxa and flagellated taxa become less abundant, and slower-growing taxa (i.e.,

taxa with fewer 16S rRNA gene copies) and sporulating taxa become more abundant. Intriguingly, during this time, microbiomes become more similar across infants in both taxonomic and trait-based compositions. Taxonomic turnover continues after the first year, but is largely redundant with respect to the traits examined. The trait-based patterns in our analysis suggest that succession begins with a functionally variable cohort of early arrivers, adept at proliferating rapidly within hosts, which gradually matures into a more functionally uniform cohort of taxa able to both thrive in the anoxic gut environment and disperse between anoxic patches (e.g., guts) as oxygen-tolerant spores.

## Results

**Trait-based patterns of succession.** We observed consistent taxonomic and trait-based shifts in infant gut microbiomes during the first 3 years of infant life (Fig. 1, Fig. 2). With respect to taxonomic composition, early succession was dominated by Bacteroidaceae and Bifidobacteriaceae (Fig. 1a, b), whereas late succession was dominated by Lachnospiraceae, Ruminococcaceae, and (still) Bacteroidaceae (Fig. 1e, f). About three-fourths of the operational taxonomic units (OTUs) in this study, defined using a threshold of 97% sequence similarity in the 16S rRNA V4 region, exhibited significant positive or negative trends in abundance over succession across all infants, based on linear regressions. The extensive number of significant trends emphasizes the taxonomically predictable nature of gut microbiome development. To evaluate trait-based shifts over development, we combine curated trait data and hidden state predictions to generate a custom database of 12 microbial traits for the OTUs in the infant microbiome samples (see Methods). Early and late successional specialists differed significantly in their predicted trait values: late successional specialists were less tolerant of oxygen, were more capable of sporulation, and had higher temperature optima than early successional specialists (Supplementary Figure 1).

Community weighted means (CWMs) of several traits trended significantly over the course of succession (Fig. 2), illustrating the functionally predictable nature of gut microbiome development[30]. A CWM is the mean trait value of the OTUs in a community, weighted by their relative abundances. Ecologically speaking, CWMs characterize the dominant traits of a community, and can be thought of both in terms of how they reflect system properties (i.e., as response traits) and how they influence system properties (i.e., as effect traits)[31]. For example, oxygen-tolerant taxa (e.g., facultative anaerobes) present at the onset of succession were rapidly overtaken by obligate anaerobes (Fig. 2i), presumably in response to a drop in gut oxygen concentration due to increased uptake by epithelial cells[32]. Meanwhile, the mean number of B-vitamin pathways in OTU genomes decreased over time (Fig. 2b), contradicting our expectation that human hosts would selectively enrich such taxa over the course of succession to promote the production of these essential nutrients.

Pronounced shifts in the predicted values of two traits potentially related to dispersal ability suggest that dispersal dynamics may play a key role in shaping successional patterns. First, the initial presence and subsequent decline of taxa likely to have flagella (Fig. 2h) could mean that the ability to actively disperse over short distances (i.e., spread within hosts) improves colonization rates during early succession, but that flagella are not as advantageous in the mature gut. In support of this, unflagellated strains have been shown to be poorer colonizers of chickens' gastrointestinal tracts than flagellated strains[33], and a positive relationship has been drawn between motility and bacterial transmission[34]. Second, the increase over time in predicted sporulating ability (Fig. 2j, Supplementary Figure 3) may reflect the long-term advantages of being able to disperse

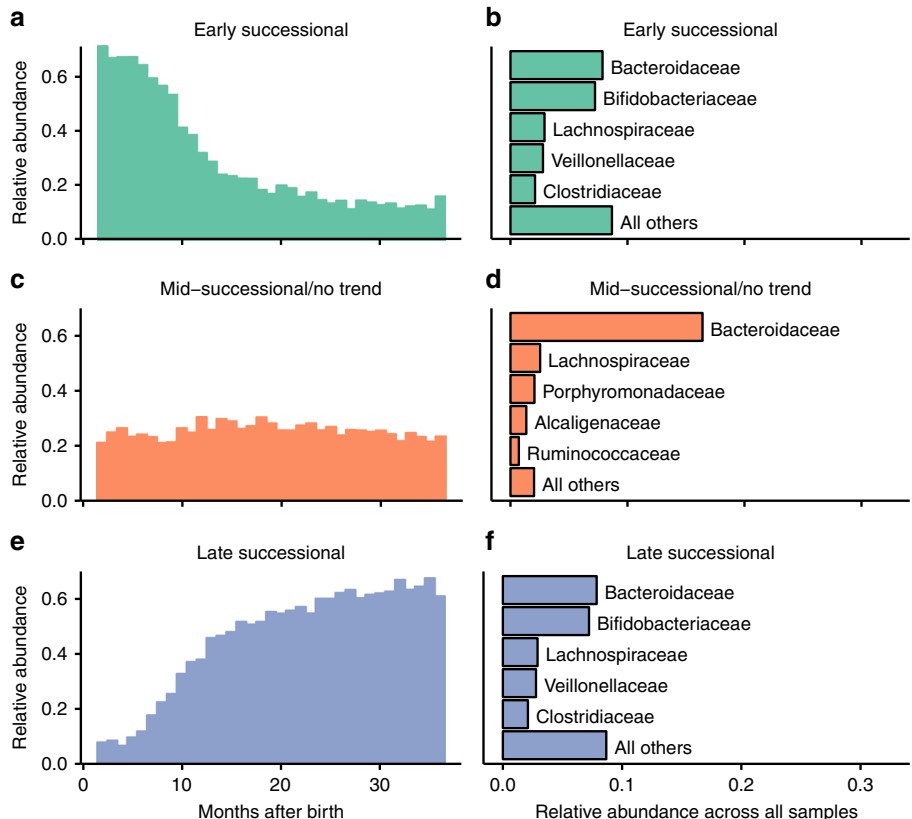

**Fig. 1** OTU abundances over gut succession. OTUs were placed into one of three successional groups based on their average trends in abundance across 56 infants over development. Taxa were categorized as early successional if their abundances increased significantly over time ($p < 0.05$), and late successional if their abundances decreased significantly over time, based on linear regressions; OTUs that did not trend significantly over time were placed into a mid-successional or no-trend category. **a**, **c**, and **e** Combined relative abundances of the OTUs of each successional category over time. **b**, **d**, and **f** The five most commonly-represented families among the OTUs of each successional category, and their total relative abundances over the entire sampling period

among hosts and/or persist within hosts in a dormant state during stressful conditions[24,35]. As succession proceeds and the gut environment becomes increasingly anoxic, obligate anaerobes gain a competitive advantage over facultative anaerobes because they do not need to maintain the machinery for tolerating oxidative stress. However, this advantage comes at the cost of being more vulnerable to oxidative stress while dispersing through oxic environments to colonize new hosts. Sporulating taxa circumvent this potential tradeoff by traversing oxic environments as oxygen-tolerant spores, and then thriving in the gut as anaerobes. The observed increase of sporulating taxa over gut community development, both in total abundance (Fig. 2j) and OTU richness (Supplementary Figure 3), likely reflects the steady arrival and successful colonization of these taxa well-adapted for the anoxic gut environment.

The mean predicted number of 16S rRNA gene copies, a genomic trait associated with the ability to quickly exploit available resources due to higher maximum potential growth rates[36], decreased steadily in gut microbiomes over time (Fig. 2a). A decrease in mean 16S rRNA gene copy number over time is characteristic of primary succession in microbial systems that are initially rich in resources[8], such as a vial of sterile nutrient broth placed in an open-air environment[37]. However, a decrease in mean 16S rRNA gene copy number could also arise if faster-growing taxa thrive on easily-digested milk or formula, the primary carbon source during early succession, and slower-growing taxa only begin to thrive as the primary carbon source shifts toward increasingly complex molecules derived from solid food. In either case, the decrease in mean 16S rRNA gene copy number over time likely reflects a shift from taxa capable of rapid low-efficiency growth to slower high-efficiency growth over succession[23,38].

Many predicted traits correlated significantly among taxa (Supplementary Figure 2). The strongest positive correlations were between gene number and genome size, genome size and B-vitamin pathway number, and sporulation and Gram-positive status, while the strongest negative correlations were between optimal growth temperature and oxygen tolerance, Gram-positive status and B-vitamin pathway number, and GC content and 16S rRNA gene copy number. The remaining Pearson correlation coefficients were less than 0.6 or greater than −0.6. On one hand, correlations among traits are noteworthy because they may be independent indicators of a taxon's position on the same ecological tradeoff axis (i.e., they may constitute a trait syndrome). For example, the negative correlation observed between sporulation score and oxygen tolerance may be because these traits provide two alternative methods of dealing with oxidative stress, either by becoming metabolically dormant until oxidative stress is relaxed, or by carrying the cellular machinery to tolerate it, respectively. On the other hand, correlations among traits may simply be artifacts of arbitrary genomic linkage, and not independent instances of evolutionary adaptation. As such, the mechanisms we invoke as possible explanations for the trait-based patterns observed in this study are merely hypotheses which hopefully spur further experimental work.

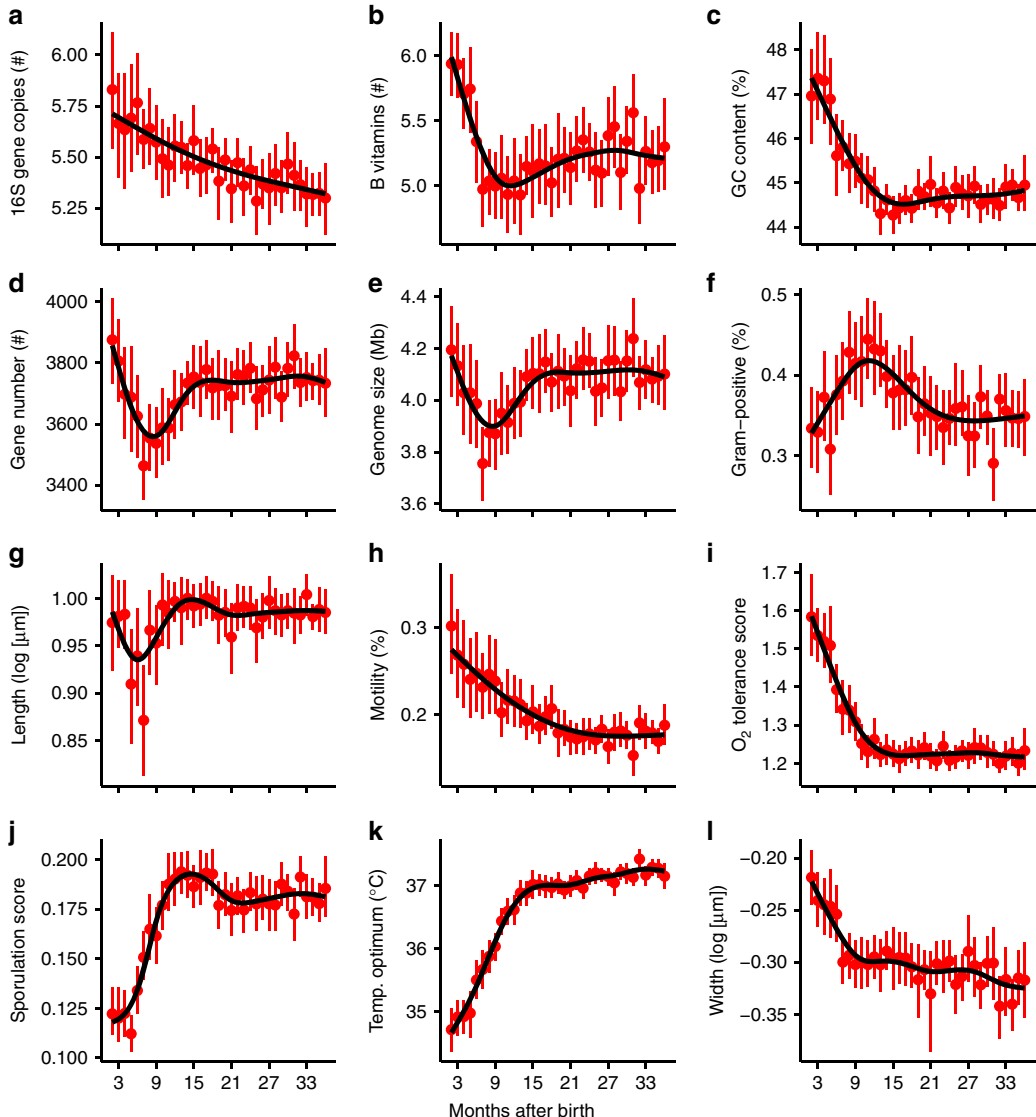

**Fig. 2** Abundance-weighted trait means over gut succession. Filled red circles show average abundance-weighted means of predicted trait values of gut microbiomes of up to 56 infants in each month of development (**a-l**). N is equal to the number of samples in each month, and ranges from 27 to 59. Vertical red lines show 95% confidence intervals. Black trendlines were fit using generalized additive models

To explore how early arrival of different taxa could affect the trajectory of gut succession, we compared trait-based successional patterns of infants delivered vaginally and by C-section (Fig. 3). We reasoned that any consistent community differences between the two groups of infants would likely arise due to differences in early colonization, i.e., because infants born vaginally were initially colonized by taxa from the mother during delivery, and infants born by C-section were initially colonized by a different cohort of taxa arriving from the ambient environment (e.g., the mother's skin, hospital surfaces). Notable trait-based differences between the microbiomes in C-section infants, relative to those in vaginally delivered infants, were initially elevated numbers of Gram-positive taxa (Fig. 3f), and prolonged persistence of oxygen-tolerant taxa (Fig. 3i). There were also initially elevated mean 16S rRNA gene copy numbers (Fig. 3a) and initially higher prevalence of flagellated taxa (Fig. 3h) in C-section infants, relative to vaginally born infants, but these differences were not statistically significant after accounting for multiple comparisons. At minimum, these results suggest that taxa encountered by infants during vaginal delivery are functionally distinct from those encountered by infants after C-section delivery in the

hospital environment. More interestingly, however, they suggest that gut colonization patterns differ depending on the composition of the initial pool of colonizing taxa. Significant trait-based compositional differences by birth mode persisted for up to 2 years (Fig. 3i), corroborating previous research showing that differences in early colonization can have lasting effects on community composition[39,40], a phenomenon also termed priority effects[41,42]. On the other hand, sustained trait-based differences between infants by delivery mode are surprising, given recent work that found strong selective forces to quickly discourage the growth of immigrant taxa from the mother's skin or birth canal;[43] hence, our findings suggest that the persistent differences by birth mode may result from a lack of arrival (i.e., dispersal limitation) of gut-adapted taxa from the mother, rather than qualitatively different community filters among infants.

Exposure to antibiotics was associated with consistent trait-based shifts in gut microbiome composition (Fig. 3). Specifically, infants exposed to repeated antibiotic treatments had gut taxa that were on average less likely to be Gram-positive (Fig. 3f), smaller (Fig. 3g), and less capable of sporulation (Fig. 3j) than infants exposed to no antibiotics. Decreases in the relative

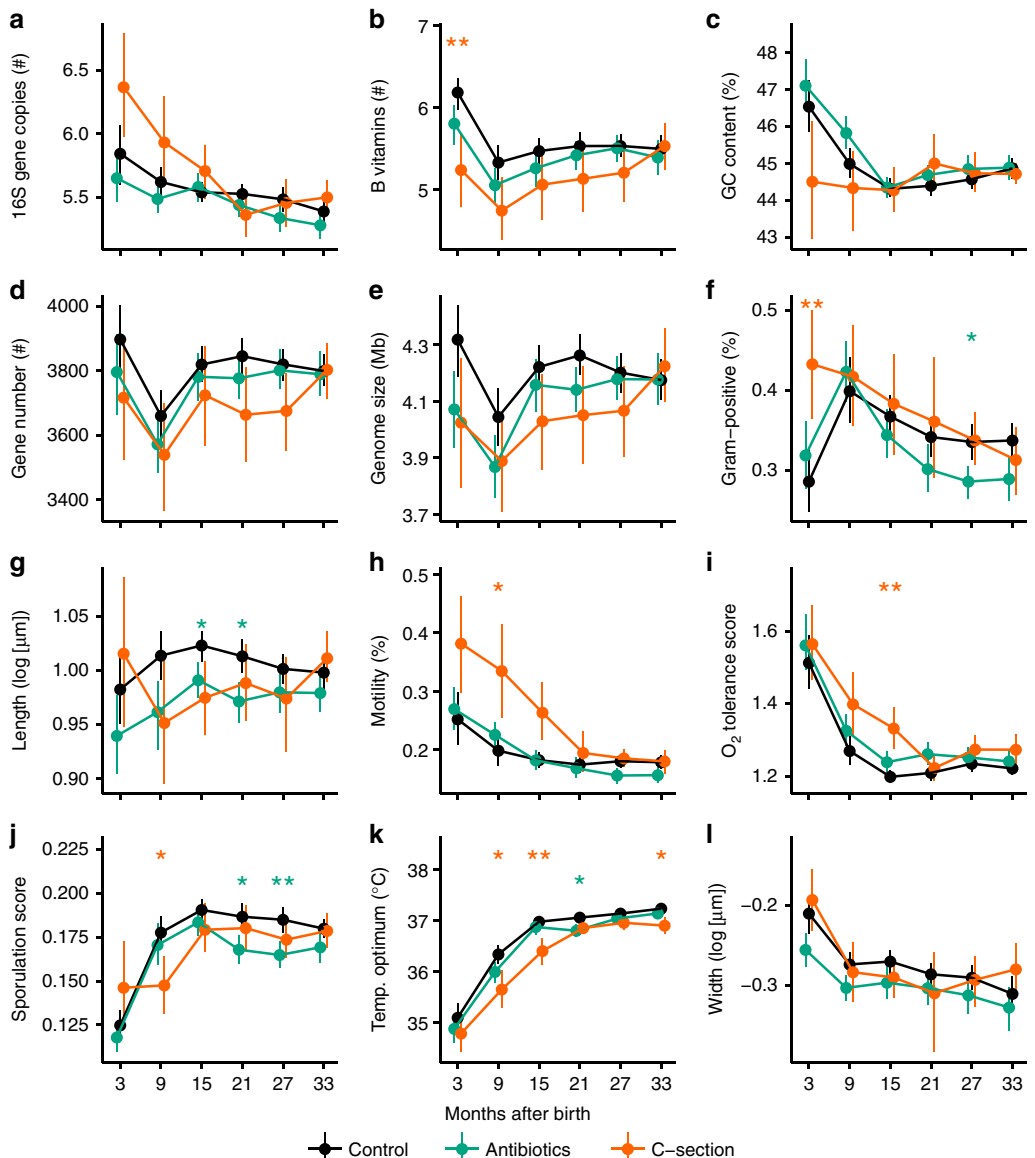

**Fig. 3** Trait-based successional patterns differ by delivery mode and antibiotic history. Abundance-weighted means of predicted trait values of infant microbiomes over succession, grouped by infant delivery mode and antibiotic history (**a-l**). Filled circles show average abundance-weighted trait means of samples within 6-month periods in each cohort of infants. N is equal to the number of samples in each 6-month period; there were between 25 to 31 total samples from six infants delivered by C-section who received little to no antibiotics ("C-section" group), 66–91 total samples from 18 infants who were treated with antibiotics for at least 50 days ("Antibiotics" group), and 72–93 total samples from 18 control infants that were delivered vaginally and received no antibiotics ("Control" group). Vertical lines show 95% confidence intervals. Asterisks denote significance based on Welch t tests performed between each treatment group and the control group (*: adjusted $p < 0.05$; **: adjusted $p < 0.01$; ***: adjusted $p < 0.001$)

abundances of Gram-positive taxa over time is arguably expected given that Gram-positive taxa lack the protective outer membrane that make Gram-negative bacteria generally more resistant to antibiotics[44]. The drop in mean predicted sporulation score is less expected, given that spores are generally very resistant to antibiotics[24]. However, spore formation is far from the only mechanism of antibiotic tolerance in Bacteria, and other strategies may be more effective for survival in the gut environment. For instance, antibiotic treatments usually result in decreases in the relative abundances of spore-forming taxa in the class Clostridia, and increases in the relative abundances of non-spore-forming taxa in the family Enterobacteriaceae[32]. More generally, consistent with prior work[45], the persistent differences in trait-based community composition between infants that underwent heavy antibiotic treatments and those that did not suggests that these

disturbances can exert long-term effects on community structure and function.

Trait variances within infant gut communities decreased over time in seven traits, and increased over time only in three traits (Supplementary Figure 4). The overall decrease in trait-based variance over time indicates that individuals of the gut community became more functionally homogeneous with respect to the traits examined in this study, perhaps due to increasingly strict environmental filtering processes[46] and/or competitive exclusion of poorly adapted taxa[47].

**Comparing taxonomic and trait-based successional patterns.** To evaluate the degree to which taxonomic changes aligned with trait-based changes, we compared taxonomic and trait-based

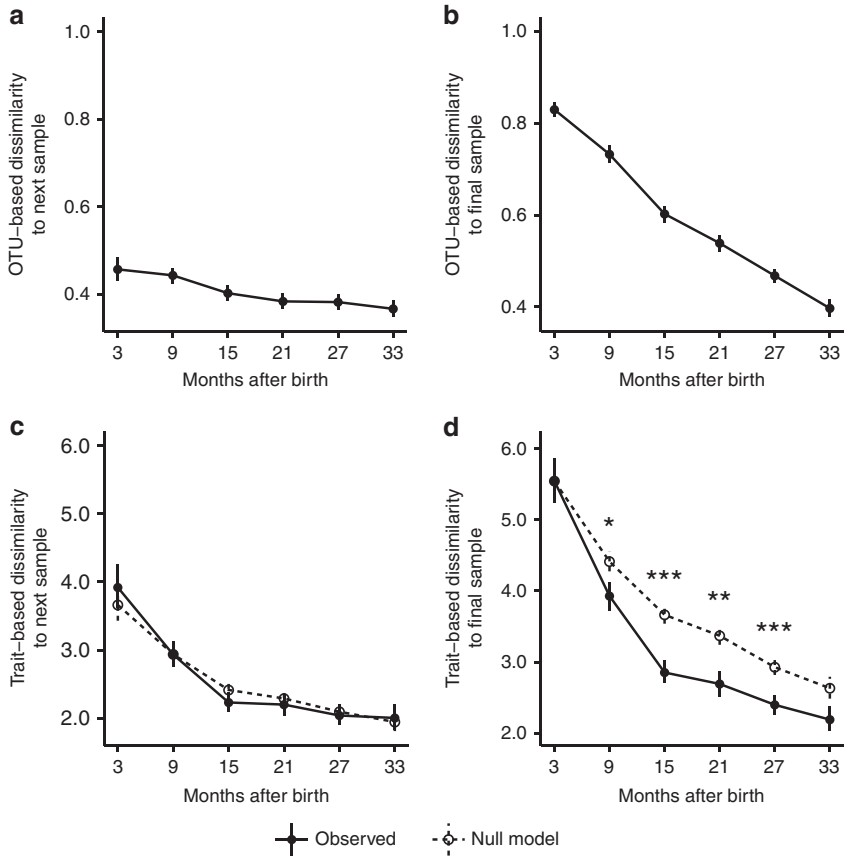

**Fig. 4** Trait-based composition stabilizes earlier than taxonomic composition. Filled circles show mean pairwise compositional dissimilarities of gut microbiome samples collected from individual infants, averaged within 6-month periods for each infant, and then across infants. OTU-based dissimilarity was calculated using Bray–Curtis dissimilarity. Trait-based dissimilarity was calculated using multidimensional Euclidean distance after scaling the distributions of values for each trait to ensure equal contribution. **a** Mean OTU-based dissimilarity between subsequent samples declines slightly over time. **b** Mean OTU-based dissimilarity between samples and the final samples taken from each infant decreases steadily throughout the sampling period until finally reaching baseline levels of between-sample dissimilarity in the last 6-month period, seen in **a**. **c** Mean trait-based dissimilarity between subsequent samples appears elevated in the first year, but does not differ significantly from null model predictions that assume trait-agnostic turnover (see Methods). **d** Mean trait-based dissimilarity between samples and the final samples taken from each infant decreases rapidly and approaches baseline levels of between-sample dissimilarity within the first year, seen in **c**. Moreover, trait-based community composition converges toward that of the final sample significantly faster than null model expectations, illustrating the non-random nature of trait-based community turnover over succession. In all panels, $N$ equals 56, the number of infants. Vertical lines show 95% confidence intervals. Asterisks denote significance between observed and null model predictions based on Welch $t$ tests (*$p < 0.05$; **$p < 0.01$; ***$p < 0.001$)

turnover over time within infants, both in terms of short-term compositional variability (measured as the dissimilarity between subsequent samples) and long-term directional turnover (measured as the dissimilarity between each sample and the final sample collected). Compositional variability was higher in the first year of development, both in terms of OTUs (Fig. 4a) and predicted traits (Fig. 4c), than in the second or third years of development. A decrease in compositional variability over time is a classical feature of many ecological successional systems[48]. To evaluate whether trait-based compositional variability was higher or lower than expected by chance, given the magnitudes of taxonomic variability observed, we compared observed patterns to predictions from null model simulations for which trait values were randomly shuffled among taxa and trait-based compositional variability was re-calculated (see Methods). In other words, we calculated what trait-based compositional variability would look like if the traits in our study were completely decoupled from taxon performance. Differences between observed and null model predictions were neither large nor significant (Fig. 4c), suggesting that the traits in our study had little influence on compositional variability over succession.

An analysis of directional turnover over succession revealed that infant gut communities matured and stabilized faster in their trait-based compositions than in their OTU-based compositions. Specifically, OTU-based directional turnover was relatively steady across all 3 years of study (Fig. 4b), whereas trait-based directional turnover was high only in the first year (Fig. 4d) before dropping to nearly-baseline levels of trait-based compositional variability (Fig. 4c). Trait-based directional turnover significantly exceeded null model predictions of trait-agnostic turnover (Fig. 4d), suggesting that infant gut microbiomes stabilize (i.e., cease to exhibit directional turnover) in terms of traits and their associated functions sooner than they stabilize in terms of OTUs, aligning with previous metagenomic work[30]. The fact that OTU-based directional turnover continued steadily over the first 3 years of infant development despite a relative slowing of trait-based directional turnover after the first year indicates that late-stage OTU-based turnover was functionally redundant with respect to the traits examined in this study. Functionally redundant turnover could arise due to variable immigration rates (i.e., if functionally redundant taxa immigrated into the gut at variable rates over time), or due to ecological drift (i.e., if

functionally redundant taxa increased or decreased in relative abundances due to stochastic birth/death events). With respect to the latter: even though the gut community has a large number of individuals (i.e., cells), which, all else being equal, makes it less susceptible to ecological drift[49], many of its constituent taxa are rare and therefore still vulnerable to stochastic variation in their relative population sizes over time. Future work should quantify immigration rates, and consider other traits as potential drivers of late-stage successional community turnover, such as those relating to metabolism of specific dietary compounds[50], cross-feeding[6], or phage-host interactions[51].

**Compositional differences across microbiomes**. Surprisingly, gut community compositions became more similar (i.e., converged) across infants as they matured (Fig. 5). This ran counter to our expectations that gut community compositions would diverge as infants shifted from subsisting on milk and/or formula (i.e., simple substrates with low resource variability expected among hosts) to solid foods (i.e., complex substrates with higher resource variability expected among hosts), and as interactions between infants and their idiosyncratic home environments accumulated over time. Compositional convergence across infants over development may reflect a process whereby a stochastic cohort of initial taxa colonize infant guts but are gradually replaced, or supplemented with, taxa better suited for the gut environment. Such initial compositional differences among infants could be generated by stochastic colonization dynamics, differences in the pool of potential immigrants from the infants' mothers, or a combination of the both. Regardless, it is likely that gut community convergence across infants over development is partly due to the delayed arrival of taxa well-adapted for the gut environment, i.e., dispersal limitation.

Compositional convergence among infant gut communities was more pronounced and abrupt in terms of traits (Fig. 5b) than in OTUs (Fig. 5a), which converged only slightly and gradually over time. Trait-based rates of convergence significantly exceeded null model expectations of trait-agnostic convergence (Fig. 5b), indicating that trait-based convergence was not random with

respect to the traits examined in this study. This discrepancy between OTU-based and trait-based patterns of convergence among infants leads to two insights. First, it is another reminder that microbial communities with different OTU-based compositions do not necessarily differ in their functional potentials[30,52]. Second, it means that community succession can be more predictable with respect to traits than OTUs. Once again, these results indicate that OTU-based turnover over late succession is largely functionally redundant with respect to the traits examined. Functional redundancy among gut microbiome taxa may benefit the host by improving community resilience in response to disturbance[53]. Interestingly, mean compositional differences among infants born by C-section were, on average, greater both in terms of OTU-based and trait-based dissimilarity (Supplementary Figure 5). Such differences could arise if the taxa to which C-section infants are initially exposed are more taxonomically and functionally variable than the taxa to which vaginally delivered infants are exposed.

## Discussion
As in the ecological studies of macroorganisms, trait-based analysis of gut microbiome succession offers insights into the mechanisms of community assembly, such as dispersal limitation and ecological filtering, and the balance between stochastic and deterministic forces. The stabilization of trait-based community composition after the first year of development (Fig. 4), and the drop in variance of predicted trait values in gut communities for most traits over time (Supplementary Figure 4), both suggest that succession is at least partially functionally deterministic, with early dynamics potentially reflecting stochastic colonization during the birthing process, followed by the gradual colonization and enrichment of a more functionally uniform cohort of taxa better adapted for the mature gut environment. Rates of OTU-based directional turnover remained steady over the first 3 years of succession (Fig. 4b), even though trait-based directional turnover essentially stabilized after only 1 year (Fig. 4d), underscoring the fact that OTU-based compositional changes need not imply changes in trait-based composition[54]. However, there are

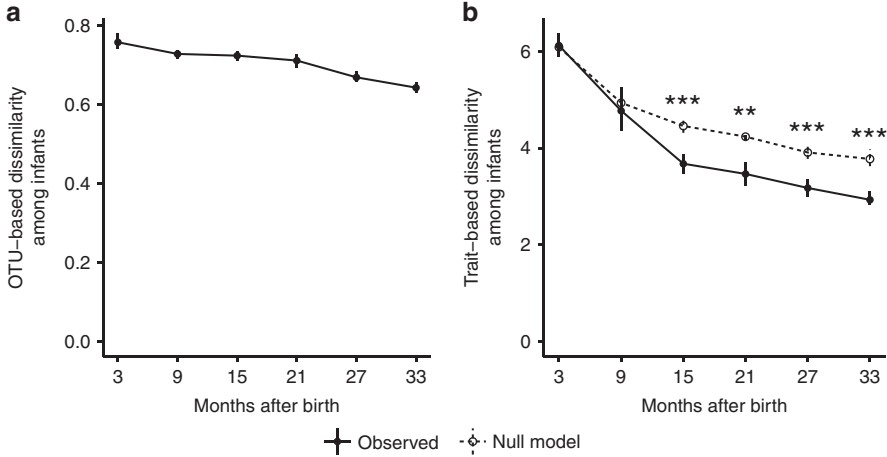

**Fig. 5** Infants' microbiomes converge compositionally over time. Filled circles show mean compositional dissimilarities of gut microbiomes across infants within each 6-month periods. Mean dissimilarities were calculated by first taking the mean dissimilarity of all sample pairs, except those from the same infant, in each of the first 36 months of development (for these means, *N* ranges from 88 to 3410), and then taking their means within each 6-month period; hence, for each circle, *N* equals 6. OTU-based dissimilarity was calculated using Bray–Curtis dissimilarity. Trait-based dissimilarity was calculated using multidimensional Euclidean distance after scaling the distributions of values for each trait to ensure equal contribution. **a** OTU-based dissimilarity among infants decreased slightly over time, indicating a modest convergence in taxonomic composition. **b** Trait-based dissimilarity among infants fell quickly over the first 18 months and then remained relatively static thereafter, indicating rapid convergence in trait-based composition during early succession. The magnitude of trait-based compositional convergence across infants was significantly greater than predicted by a null model assuming trait-agnostic turnover. Vertical lines show 95% confidence intervals. Asterisks denote significance between observed and null model predictions based on Welch *t* tests (*$p<0.05$; **$p<0.01$; ***$p<0.001$)

surely aspects of community assembly that cannot be understood using only the traits used in this study, and future work should expand the number of traits considered. Moreover, because our study is observational, we cannot distinguish between an OTU that fails to disperse to a potential host and an OTU that arrives but fails to establish, so future research should also explore the relationship between OTU arrival and detection in fecal samples to better disentangle dispersal limitation and niched-based differences among taxa.

Comparisons of trait-based patterns between cohorts of infants are an opportunity to understand the effects of specific events (e.g., delivery mode, antibiotic exposure), and serve as natural experiments that can reveal how gut communities respond to, and recover from, systematic disturbances. In our analysis, for example, delivery mode resulted in sustained differences in community composition, indicating that priority effects can play an important role in gut community assembly[41,42], a result that likely extends to other types of disturbance during early life, such as gastrointestinal illness or malnutrition. Similarly, repeated antibiotics treatments led to significant differences in trait-based community compositions (Fig. 3), suggesting that gut communities are not infinitely functionally resistant and/or that tradeoffs exist between antibiotic resistance and other traits[52]. Understanding trait-based differences between other cohorts, such as healthy versus diseased[55], or on and off specific diets[56], could provide insight into additional factors shaping gut microbiome community assembly. For example, the unhealthy, dysbiotic gut may have a higher prevalence of microaerobic and biofilm-forming species[57], a difference that could be detected using trait-based analyses. Trait-based approaches, which link organismal structures to ecological functions, are poised to advance our mechanistic understanding of the gut microbiome, and their usefulness will only increase as we improve our knowledge of how traits mediate microbial interactions and as we increase the depth and breadth of microbial trait databases.

## Methods

**Infant microbiome sampling and sequence processing**. Our foremost aim in this study was to characterize general patterns of gut primary succession that hold true regardless of host-related differences. As such, unless otherwise noted, we include all infants in our analyses, regardless of delivery mode or other host differences specific to each included study. Longitudinal infant gut microbiome data were compiled from two studies from the DIABIMMUNE study group (https://pubs.broadinstitute.org/diabimmune), one focused on the effects of antibiotics on gut community development[28], and the other focused on the effects of type-1 diabetes on gut community development[27]. In the antibiotics study, infants either had nine or more antibiotic on gut community development courses, or no antibiotic courses[28]. In the type-1 diabetes study, infants tested positive for HLA DR-DQ alleles conferring risk of type-1 diabetes; of the infants which met our sampling criteria (see below), three developed type-1 diabetes during the sampling period[27].

Stool samples of infants were collected by participants' parents and stored in their house freezers until the next scheduled visit to the local study center. Samples were then shipped on dry ice to the DIABIMMUNE Core Laboratory, where they were stored at −80 °C until being sent to the Broad Institute for DNA extraction and 16S rRNA amplicon sequencing. Sequencing was performed on the Illumina HiSeq 2500 platform using the 515 F and 806 R primers. Of 74 infants across the two studies, only those with at least 12 samples and those which extended more than 30 months were used in this study, yielding 56 infants with 12–36 sampling points (mean = 26.45; median = 27) taken at semi-regular intervals over the first 3 years of infant life (Supplementary Figure 6). All subjects were from Finland, except one from Estonia.

Infants varied in their modes of delivery and antibiotic histories, providing an opportunity to explore the potential effects of these natural experiments on trait-based gut community composition. To this end, infants were divided into three groups: (1) high antibiotic exposure ($n = 18$), if they underwent at least 50 days of antibiotic treatment and were delivered vaginally, (2) C-section delivery ($n = 6$), if they were delivered by C-section and underwent two or fewer rounds of antibiotics, and (3) a control group that was delivered vaginally and received no antibiotic treatments ($n = 18$). In some instances, antibiotic treatment durations were not reported, in which case we assumed 7 days per treatment. Twelve types of antibiotics were administered for a variety of ailments, with the most common being amoxicillin, trimethoprim, and sulfadiazine aimed at treating acute ear

infections. Infant metadata, drawn from the two studies from which sequence data for this study are drawn[27,28], is available in Supplementary Data 1.

Sequence processing was done using USEARCH version 10.0.240[58]. Raw sequencing data were downloaded from the DIABIMMUNE website https://pubs.broadinstitute.org/diabimmune. Chimeras and reads flagged with more than one error were excluded, and the remaining reads were truncated to 250 bp, the expected overlap when using 515 F and 806 R primers. Reads were clustered into OTUs at 97% sequence identity using the UPARSE-OTU algorithm (Supplementary Data 2). Representative sequences from each OTU were mapped to the SILVA v123 database[59] to determine potential taxonomic identities (Supplementary Data 3). To avoid bias in sampling effort, samples were rarefied to 5,000 sequences, and seven samples with fewer than 5000 sequences were removed.

**Assembling trait data**. We compiled data on 16 genomic, physiological, and life history traits of bacteria from public databases and individual studies (Table 1, Supplementary Data 4). Trait data were only included if they were explicitly associated with taxa with full Latin binomials (i.e., Genus and Species labels) and also appeared either in our SILVA-derived taxonomy file for the combined gut community samples or in the curated taxonomy file from the 132 release of the Living Tree Project[29]. Altogether, these amounted to 57,543 collected trait data spread across 10,906 taxa. When a taxon had more than one trait value, the mean or mode was used, depending on whether the trait was quantified continuously or discretely.

Descriptions and data sources for each trait are listed briefly in Table 1, but here we elaborate with a few additional details: (1) the numbers of B-vitamin synthesis pathways in the genome were drawn from ref. [60] and are based on genome annotations from the pubSEED platform[61]. (2) In some cases, optimal temperature was calculated as the mean of lower and upper temperature ranges, consistent with ref. [26]. (3) IgA binding affinity refers to the degree that immunoglobulin A bound to specific bacterial taxa, and was quantified using an IgA coating index calculated in ref. [62] using flow-cytometry-based bacterial cell sorting and 16S rRNA sequencing to characterize the coating load of IgA on specific taxa from fecal samples in a murine model. (4) Sporulation score indicates the tendency of taxa to sporulate, and was calculated in ref. [24] as a continuous score ranging from zero to one that depended on a combination of targeted phenotypic culturing and whole-genome sequencing from stool samples. When possible, we used sporulation scores from ref. [24]. When sporulation scores from ref. [24] were unavailable for a given Latin binomial, we drew on sporulation data from other repositories (Table 1), which were generally binary, either noting the presence or absence of spores; in cases when spores were present, taxa were given sporulation scores of 0.549, equal to the median sporulation score of the taxa with sporulation scores greater than zero in ref. [24]; when spores were not observed, taxa were given sporulation scores of zero.

**Predicting unknown trait data**. We estimated unknown trait data using hidden state prediction methods based on phylogenetic inference (Supplementary Data 5). Specifically, we generated a phylogenetic tree with the 3311 OTUs from our USEARCH pipeline (before any taxa were lost due to rarefying) and the 13,900 OTUs from the 132 release of the Living Tree Project (LTP)[29] (Supplementary Figure 7; Supplementary Data 6). The topology of the tree reflects percent sequence similarity among taxa in the 16S rRNA V4 region, and was generated using agglomerative clustering of a distance matrix based on the U-sort heuristic[58]. Because LTP representative sequences were of the entire 16S rRNA gene (i.e., the ribosomal small subunit), they were truncated to the 250 bp of the V4 region using 515 F and 806 R primer sequences before generating the distance matrix. Trait data were then mapped onto the tips of the phylogenetic tree with Latin binomials. The LTP database was uniquely well-suited to interface with literature-derived trait data because each sequence represents a type strain with Genus and Species annotations drawn from the literature, not inferred phylogenetically.

Missing trait values were estimated using three hidden state prediction algorithms: independent contrasts, subtree averaging, and weighted squared-change parsimony, each calculated using the R package Castor version 1.3.4[63]. The three methods have different strengths and weaknesses[63,64], but our predictions correlated strongly (Supplementary Table 1), lending confidence to our results. We ultimately used weighted square-change parsimony for our analysis, which recursively calculates locally parsimonious states for each node based on its descending subtree, until reaching a parsimonious state estimate for the tree root[65]. Because all curated trait data were either numeric or converted to numeric (e.g., Gram-negative = 0 and Gram-positive = 1), state predictions for discrete traits could be fractional (e.g., a Gram-positive score of 0.5), reflecting their probabilistic uncertainty.

Methods of hidden state prediction offer estimates for all taxa with hidden states, even when there is not sufficient confidence to warrant estimation. To mitigate this, we discarded predictions that were statistically no better than random. More specifically, for each trait, we first pruned the full phylogenetic tree so that only OTUs (i.e., tree tips) with direct trait observations remained. Next, we calculated differences in trait values for up to 10,000 randomly selected OTU pairs within each 0.005 increment of phylogenetic distance (i.e., percent 16S rRNA V4 sequence similarity). Five generic models were then used to predict average trait differences between OTU pairs, $|y|$, as a function of phylogenetic distance, $x$, and

**Table 1 Sources of trait data gathered in this study**

| Trait | Description/units | Sources |
|---|---|---|
| Aggregation score | 0 (never) to 1 (observed aggregation) | BacDive[25]; IJSEM[26] |
| B vitamins | No. B-vitamin pathways in genome | Ref. [60] |
| 16S gene copies | No. in 16S rRNA gene copies in genome | rrnDB[68] |
| GC content | Percent guanine and cytosine in genome | IJSEM[26]; NCBI[69] |
| Gene number | No. genes in genome | NCBI[69] |
| Genome size | Genome size in megabases | NCBI[69] |
| Gram-positive | 0 (Gram-negative) to 1 (Gram-positive) | BacDive[25]; GOLD[70]; IJSEM[26] |
| IgA binding affinity | log ([IgA+]/[IgA−] + 1) | Ref. [62] |
| Length | log (μm) | BacDive[25]; GOLD[70]; IJSEM[26] |
| Motility | 0 (never motile) to 1 (always motile) | BacDive[25]; GOLD[70]; IJSEM[26] |
| Oxygen tolerance | 0 (obligate anaerobe) to 5 (obligate aerobe) | BacDive[25]; GOLD[70]; IJSEM[26] |
| pH optimum | pH | GOLD[70]; IJSEM[26] |
| Salt optimum | g per l | IJSEM[26] |
| Sporulation score | 0 (never sporulates) to 1 (sporulates easily) | BacDive[25]; GOLD[70]; IJSEM[26]; ref. [24] |
| Temperature optimum | °C | IJSEM[26] |
| Width | log (μm) | BacDive; GOLD[70]; IJSEM[26] |

IgA: immunoglobulin A, BacDive: bacterial diversity metadatabase, IJSEM: International Journal of Systematic and Evolutionary Microbiology, GOLD: Genomes OnLine Database: Joint Genome Institute, NCBI: National Center for Biotechnology Information, rrnDB: the ribosomal RNA operon copy number database

the best fitting model was selected by AIC. The models included: (1) Null: $|y| \sim 1$; (2) Linear regression: $|y| \sim x$; (3) Logarithmic regression: $|y| \sim \log(x)$; (4) Asymptotic regression: $|y| \sim a(1 - e^{(-e^{b}x)})$, where $a$ and $b$ were determined using a self-starting nonlinear least squares approach, and the model fit was constrained to pass through the origin; and (5) Logistic regression: $|y| \sim \frac{a}{1 + e^{(\frac{b-x}{c})}}$, where $a$, $b$, and $c$ were determined using a self-starting nonlinear least squares approach. Null models provided the best fit for aggregation score, IgA binding affinity, pH optimum, and salt optimum, indicating that for these traits, trait values should not be estimated at any phylogenetic distance. For the remaining 12 traits, we identified the phylogenetic distances at which the values of each trait were no longer evolutionarily conserved, i.e., when model-predicted trait differences between OTU pairs were no different than null expectations. We defined null expectations as the mean trait difference of all OTU pairs with more than 0.1 phylogenetic distance between them. We only predicted traits of OTUs when there were taxa with known (i.e., literature-derived) trait values within trait-specific thresholds of phylogenetic distance; we defined these thresholds as the points at which model predictions rose to 90% of null expectations (Table 2; refer to Supplementary Figure 8 for a graphical rendering of the approach). Of the traits that were amenable to hidden state prediction, coverage ranged from 78.7% (16S rRNA gene copy number) to 99.9% (temperature optimum) of sequences used in this study (Supplementary Figure 9). We assessed statistical independence among traits predictions using Pearson correlation coefficients; $p$-values were adjusted for multiple comparisons using the Benjamini–Hochberg procedure.

**Trait-based successional patterns within and across infants**. Trait-based successional patterns were evaluated at both the OTU-level and the community level (i.e., on the level of individual samples). For the OTU-level analysis, OTUs were placed into one of three groups based on results of linear models of OTU abundances over time across all infants: OTUs with significant negative trends in abundance over time ($p < 0.05$, $\beta < 0$) were categorized as early successional; OTUs with significant positive trends in abundance over time ($p < 0.05$, $\beta > 0$) were categorized as late successional; otherwise, taxa were placed into a third category that included OTUs with sporadic, unvarying, or hump-shaped patterns of abundance over time. Statistical differences in the predicted trait values of OTUs in the three groups were evaluated with Welch $t$ tests; $p$-values were adjusted for multiple comparisons using the Benjamini–Hochberg procedure.

Trait-based differences at the community level were quantified using CWMs. A CWM is the mean trait value of the OTUs in a community, weighted by their relative abundances. Here, a CWM is formally equal to $\sum_{i=1}^{S} p_i x_i$, where $p_i$ is the abundance of OTU $i$ ($i = 1, 2, \dots S$), and $x_i$ is the trait value for OTU $i$. We used Welch $t$ tests to test for differences in CWMs between infants treated with and without antibiotics, and infants delivered by C-section and vaginally, for each 6-month period of infant development; $p$-values were adjusted for multiple comparisons using the Benjamini–Hochberg procedure.

**Comparison of taxonomic and trait-based turnover**. We quantified differences in microbiome community compositions in two ways. First, we used Bray–Curtis dissimilarity to quantify differences in the OTU-based compositions of samples[66]. Second, we quantified trait-based differences among communities with multi-dimensional Euclidean distance[67]. Specifically, Euclidean distance between two communities was calculated by (1) scaling predicted trait values by their standard

**Table 2 Maximum phylogenetic distances used to infer trait values**

| Trait | Max. distance |
|---|---|
| Aggregation score | 0.00 |
| B vitamins | 0.06 |
| 16S gene copies | 0.05 |
| GC content | 0.12 |
| Gene number | 0.08 |
| Genome size | 0.09 |
| Gram-positive | 0.10 |
| IgA binding affinity | 0.00 |
| Length | 0.12 |
| Motility | 0.08 |
| Oxygen tolerance | 0.11 |
| pH optimum | 0.00 |
| Salt optimum | 0.00 |
| Sporulation score | 0.06 |
| Temperature optimum | 0.14 |
| Width | 0.12 |

Percent sequence dissimilarities (i.e., phylogenetic distances) in the 16S rRNA V4 region at which statistical support for trait conservatism disappears for each trait (see Methods and Supplementary Figure 9)

deviations to give each trait equal weight, (2) calculating the CWMs of each trait for both communities, and then (3) using the Pythagorean theorem to determine the distance between the two communities in $n$-dimensional trait space.

We examined OTU-based and trait-based community changes over time in two ways. First, to quantify changes in short-term compositional variability over infant development, we examined compositional differences of subsequent samples from the same infant, at intervals approximately between 1 to 3 months. Second, to quantify rates of long-term directional turnover over infant development, we examined compositional differences between samples and the final sample from each infant. To determine whether trait-based rates of compositional variability and directional turnover exceeded those expected by chance, we compared observed rates of trait-based turnover to null models of trait-agnostic community change. Specifically, we generated 1000 mock versions of our data with trait values randomly shuffled among OTUs, and recalculating pairwise sample dissimilarities. In other words, null models reflect what trait-based turnover would have been if organismal traits were unrelated to performance. We tested for statistical differences between observed and null turnover rates within 6-month periods using Welch $t$ tests.

To determine if community composition converged or diverged across infants as development progressed, we divided samples into 1-month slices and calculated mean OTU-based and trait-based distances for all pairwise combinations of samples, excluding pairs of samples from the same infant. To determine whether

observed rates of trait-based compositional convergence/divergence across infants differed from those expected by chance, we compared our observations to null models of trait-agnostic community changes over time. Similar to our analysis of trait-based turnover within infants, null models were performed by randomly shuffling trait values among OTUs and recalculating pairwise sample dissimilarities. We tested for statistical differences between observed and null model rates of convergence within 6-month periods using Welch $t$ tests.

## Data availability

Raw sequencing data are available online at the NCBI project accession numbers PRJNA231909 and PRJNA290381. Custom scripts used in the bioinformatic pipeline and statistical analyses are available at: https://github.com/ShadeLab/microbiome_trait_succession. All relevant data used in this study are included as Supplementary Data files, available at: https://figshare.com/projects/Trait-based_succession_of_the_infant_gut_microbiome/58202.

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

## Acknowledgements

This work was supported in part by Michigan State University through computational resources provided by the Institute for Cyber-Enabled Research. A.S. acknowledges support from the National Science Foundation under Grant No DEB#1749544. J.G. was supported by the Michigan State University Foundation funding to E.L.

## Author contributions

J.G., A.S. and E.L. conceived the study, J.G. developed methods, analyzed data, and wrote the paper, and all authors discussed analysis and revised the paper.

## Additional information

**Competing interests:** The authors declare no competing interests.

