## [Peer Review File · Nature Communications]

Reviewers' Comments:

Reviewer #1:

Remarks to the Author:

The authors present an intriguing analysis of the taxonomic and "functional" trajectory of infant human gut microbiota, which I think will be a valuable contribution to the field. I liked the core ideas of the paper, and I sympathize with its overall conclusions. However, I have substantial reservations regarding the suitability of the presented methods, and in some cases I found that crucial analytical steps, needed to support the authors' conclusions, were missing. Fortunately, I believe that all of these limitations can be addressed, although the conclusions of the paper may change. I wish the authors all the best in improving their paper and hopefully strengthening their case.

Major comments

1. Choice of Hidden State Prediction (HSP) method: The authors' approach to predicting the trait value ("state") for uncultured OTUs was to take the average state of all cultured OTUs descending from a given node ("subtree averaging"), and then assign that average state to the node's uncultured OTUs. I should say upfront that there are much better ways to do HSP than subtree averaging, and which are not substantially computationally harder. For example, Mk models or maximum parsimony approaches for discrete states, and squared change parsimony for continuous traits, are all included in the same R package used by the authors. One issue with subtree averaging is that it only provides "local" estimates, i.e. ancestral state estimates are only based on descending tips, but not sister clades. Specifically for 16S gene copy numbers, which is a trait considered by the authors, subtree averaging is actually one of the least accurate techniques available (Louca et al. 2018, BMC Microbiome, DOI:10.1186/s40168-018-0420-9), and it is likely that this also applies to other traits. I would advise the authors to consider more sophisticated HSP methods, and preferably try out a few different ones and choose the ones that work best (if possible separately for each trait); also see my next comment.

2. Validation of HSP: The authors do not provide any validation of their HSP. If I'm reading Figure S2 correctly, then for many OTUs state predictions were based on distantly related cultured strains (i.e. only within the same family or order). Many microbial traits tend to vary at much shorter phylogenetic scales (e.g. see Martiny et al. 2013, ISME J, DOI:10.1038/ismej.2012.160), and so I'm not sure how much their predictions can be trusted. At the very least I would expect the authors to perform some type of cross-validation analysis, where predictive accuracy is evaluated as a function of phylogenetic distance (separately for each trait), for example as done by Langille et al. (2013, Nature Biotechnology, DOI:10.1038/nbt.2676). I would further advise omitting OTUs from the analysis that have no sufficiently closely related cultured strain (perhaps separately for each trait, depending on the cross-validation results, as per the authors' judgement). As it currently stands, it is hard to give trust to the majority of the authors' trait predictions. Especially for traits where the authors claim an absence of a significant trend, this could be the mere outcome of an extremely inaccurate (essentially random) state prediction.

3. Lines 270-272: Related to my previous comments, could it be that the absence of a trend in 16S gene copy numbers over time is due to the high uncertainty in 16S GCN predictions? That is, if 16S GCN prediction is essentially random (as argued by Louca et al. 2018, BMC Microbiome), this would mask any real biological signal. Again, a validation of your HSP would add confidence in your trait-based analyses.

4. Lines 243-246: It is hard to see whether trait-based patterns are really statistically independent solely based on the correlation coefficients without providing any measure of significance. For

example, Pearson correlation coefficients between -0.5 and 0.5 can still contain substantial biological/ecological information. How many of the pairwise correlations between -0.5 and 0.5 were statistically significant?

5. Meaning of ecological drift (e.g. lines 353-355 and lines 409-410): The authors should be more cautious when interpreting "randomness" as "ecological drift". Ecological drift usually refers to a very specific type of randomness, namely random fluctuations of population size due to the stochasticity of birth & death events in finite (small) populations. Ecological drift is much more relevant for larger organisms (e.g. plants) which have smaller population sizes, and is arguably much less relevant for typical microbial communities. Many of the seemingly random fluctuations observed by the authors may be due to ecological processes (other than drift) not captured by the handful of traits monitored by the authors, such as phage-host interactions.

6. Comparison of taxonomic and functional convergence (e.g. lines 380-384): The authors find that convergence over time was stronger in trait-based composition than in OTU-based composition, however the authors use different distance metrics for the two (Euclidean vs Bray-Curtis). It's a bit like comparing apples to oranges. The relationship between Euclidean and Bray-Curtis distance is non-linear, and hence it is very hard to compare compositional turnover rates between the two metrics. At the very least I would expect the authors to use the same metric for both OTU- and trait-based distance, for example by treating CWMs like OTU abundances and using weighted Bray-Curtis for CWMs and OTUs. Even better would be to try both metrics and see if the conclusions are robust with regards to the choice of metric. As I mentioned before, I sympathize with the authors' conclusions, but I don't believe they are justified by the present analyses.

7. Figure 2 - 16S gene copies: It looks like there may be a trend towards fewer 16S gene copies over time in the first few weeks, which would be consistent with standard theory. In other words, high-16S-copy strains (r-strategist) have an advantage at first, but are quickly replaced by K-strategists within just a few weeks. Hence, could it be that the time scales at which some of these traits vary are much smaller (or much longer) than the sampling resolution (or sampling duration, respectively)? I think this caveat needs to be discussed in the article.

Minor comments (i.e. can easily be addressed)

How does your work relate to that recently published by Ferretti et al? *

Perhaps a brief discussion is worthwhile.

* Ferretti et al. (2018). Mother-to-Infant Microbial Transmission from Different Body Sites Shapes the Developing Infant Gut Microbiome. *Cell Host & Microbe* 24:p133–145.e5

Please be more clear when referring to the Greengenes phylogeny (or taxonomy?). Did you use a taxonomic tree or a 16S-based phylogenetic tree (the two may not be congruent). For example, if you used a phylogenetic tree for generating Fig. S2, how did you treat nodes/subtrees that were non-monophyletic taxonomically speaking?

Fig. S3: Please include information on the statistical significance of those correlations (either in the same figure, e.g. as embedded numbers) or in another corresponding figure.

Please provide a summary table as supplemental material, listing state predictions for each Greengenes OTU as used in this study. This will be useful for future studies that may want to

reproduce your results or extend your approach to other cases. In addition, since your state predictions may contain substantial error, it is important for future investigations to be able to disentangle what's a true biological result vs an artifact of your particular HSP.

Line 161: Figure number S3 seems wrong here, as Fig. S3 shows correlation coefficients, not coverages.

Lines 256-269: I'm not sure I follow your reasoning about why sporulating taxa increase with time. If sporulation presents a mechanism for dealing with oxidative stress, why would this be more important at a latter successional stage, when the gut environment becomes less aerobic overall?

Lines 291-293: This sentence was very confusing to me. Could you improve the wording?

Line 309: Please clarify at first mention what "variance" refers to (e.g. variance across individuals of the same age).

Line 348: "Such an explanation": Please clarify what "such" refers to (daily variations in diet, or cyclical community dynamics?).

Lines 423-424 ("illustrating that gut communities are not infinitely functionally resistant"): I would add "and/or that tradeoffs exist between antibiotic resistance and other traits".

Axis of Figure 1: I recommend clarifying in the left-most Y-axis label that OTUs are ranked by *time of* their peak global abundance (this only became clear to me after reading the figure caption).

Reviewer #2:

Remarks to the Author:

In this paper entitled TRAIT-BASED COMMUNITY ASSEMBLY AND SUCCESSION IN THE INFANT GUT MICROBIOME, Guittar et al apply a trait based approach to look at how microbial traits transmission over the course of succession in the human infant gut. The study leverages data produced from two published studies connected to the DIABIMMUNE study group focused on the effects of antibiotics (Yassour et al. 2016) and type-1 diabetes (Kostic et al. 2016) on gut community succession. Overall, this paper is really very interesting. It is very well written. It has a strong conceptual basis and applying a trait based approach to understand infant microbiome succession is novel. I think that the conclusions made from the data analyses are generally insightful and well presented. Additionally, the authors do well in including discussion of findings from unrelated studies and how this work provides explanations or support to the conclusions of those other studies. I thought, however, that the paper had weaknesses in methodological description and statistical rigor as detailed below.

Major Revisions

1) The description of methods that were used to generate and process the microbiome data used in this study has some major gaps. Most notably lines 105-107 where it says "Briefly, stool samples of infants were collected by participants' parents, processed at the DIABIMMUNE Core Laboratory, and sequenced at the Broad Institute for DNA extraction on the Illumina HiSeq 2500 platform, with ~2.5 Gb of sequence per sample and 101 bp paired-end reads." Is highly inadequate and really needs to be greatly expanded. It is not even clearly stated whether this is 16S rRNA amplicon or shotgun metagenomic data. Details on DNA extraction methodology, sample collection and storage, how the sequencing was conducted, how the sequence data was bioinformatically processed to e.g. remove

noise, call OTUs, assign OTUs taxonomically are all completely absent.

a. Related to this comment on line 151, it is unclear how these "Latin binomial" assignments are being made.

2) This paper is pooling data from 2 different studies – with a focus on antibiotics and Type I diabetes. Did these studies both use the same methods throughout? What were the criteria for recruiting for these different studies – e.g. did they focus on infants with particular risk attributes? To what degree might these criteria affect the degree to which the findings here are generalizable?

3) Figure S2 shows phylogenetic distances used when inferring traits. To what extent are these traits conserved at these distances? In some cases like temperature optimum, 50% of the inferences are being made at the class level. It seems to me that there should be some sort of cut off where no inference is made because the inference is uninformative at that phylogenetic distance.

4) In general, this paper lacks much statistical analysis of reported trends which I think is a major weakness. E.g. Figure S3 could indicate which stars which correlations are statistically significant. Figure S4 could indicate which traits significantly differed across early, mid and late succession. There are some other areas noted below where I think statistics would better support their arguments.

5) The cohort's characteristics are not well described which sometimes makes their results difficult to understand. As an example, on lines 371–372, the authors make a point about infants shifting from diets subsisting only on milk to solid foods, but it is not clear to me based on their cohort description the degree to which these infants are fed breast milk versus formula.

6) It is unclear to me how the authors calculated the "mean trait value" mentioned in line 176 from the methods as written.

Minor Revisions

1) With regards to the point about a lack of a trend in 16S rRNA copy number: (lines 275-278), there does seem to be a trend that this is highest in the first 2 months of life, which could indicate that it is important in very early succession. I do feel like having some sort of statistics to make this point with more confidence would be nice though.

2) With regards to the analysis of cesarian section versus vaginal delivery (lines 282-291), what was striking to me about these data in Figure 4 is just how much more variable the traits are in C-section versus vaginal delivery. Again, might be nice to have a statistical test that indicates that variance in traits is significantly greater, but based on the depicted confidence intervals this should be the case..

3) When defining the discrete time intervals (early vs. mid vs. late) in lines 164-170 the correlations with time appear to be reversed. For example, Figure 1 shows increasing abundances over time for the late succession OTUs which would appear to be positively correlated with time.

4) Typos/grammar

a. Line 112: should be "as a potential confounding variable"

b. Line 150: "similar" should be "similarity"

c. Line 178: the abbreviation (GAMs) should be inserted here

d. Line 237: "community weighted means" should just be "CWM"

e. Line 323: "(3)" is needed before "trait based Euclidean..."

5) Line 161: Figure S3 is referenced but it should be Figure S2.

Reviewers' comments:

Reviewer #1 (Remarks to the Author):

The authors present an intriguing analysis of the taxonomic and "functional" trajectory of infant human gut microbiota, which I think will be a valuable contribution to the field. I liked the core ideas of the paper, and I sympathize with its overall conclusions. However, I have substantial reservations regarding the suitability of the presented methods, and in some cases I found that crucial analytical steps, needed to support the authors' conclusions, were missing. Fortunately, I believe that all of these limitations can be addressed, although the conclusions of the paper may change. I wish the authors all the best in improving their paper and hopefully strengthening their case.

Thank you for your enthusiasm about our paper. We have taken numerous steps to improve its analytical strength, detailed below.

Major comments

1. Choice of Hidden State Prediction (HSP) method: The authors' approach to predicting the trait value ("state") for uncultured OTUs was to take the average state of all cultured OTUs descending from a given node ("subtree averaging"), and then assign that average state to the node's uncultured OTUs. I should say upfront that there are much better ways to do HSP than subtree averaging, and which are not substantially computationally harder. For example, Mk models or maximum parsimony approaches for discrete states, and squared change parsimony for continuous traits, are all included in the same R package used by the authors. One issue with subtree averaging is that it only provides "local" estimates, i.e. ancestral state estimates are only based on descending tips, but not sister clades. Specifically for 16S

gene copy numbers, which is a trait considered by the authors, subtree averaging is actually one of the least accurate techniques available (Louca et al. 2018, BMC Microbiome, DOI:10.1186/s40168-018-0420-9), and it is likely that this also applies to other traits. I would advise the authors to consider more sophisticated HSP methods, and preferably try out a few different ones and choose the ones that work best (if possible separately for each trait); also see my next comment.

We completely agree with the stated limitations of subtree averaging, and based on your suggestion, now use weighted squared-change parsimony to predict unknown trait values. In addition, we tested and found trait predictions to be consistent among three methods of hidden state prediction methods (independent contrasts, subtree averaging, and weighted squared-change parsimony) (see Table S1). Specifically: the average Pearson's correlation coefficient between predictions, across all traits, was $r = 0.952$, with the weakest individual correlation coefficient being $r = 0.872$. We also performed a versions of our analysis using each set of predictions, and found our results to be qualitatively consistent. We believe that this analysis not only strengthens confidence in the predicted trait values for phylogenetic inference, but also provides an example workflow for other microbiologists who are also unsatisfied with commonly used but less precise methods of phylogenetic inference. Thank you for the great suggestion!

2. Validation of HSP: The authors do not provide any validation of their HSP. If I'm reading Figure S2 correctly, then for many OTUs state predictions were based on distantly related cultured strains (i.e. only within the same family or order). Many microbial traits tend to vary at much shorter phylogenetic scales (e.g. see Martiny et al. 2013, ISME J, DOI:10.1038/ismej.2012.160), and so I'm not sure how much their predictions can be trusted. At the very least I would expect the authors to perform some type of cross-validation analysis, where predictive accuracy is evaluated as a function of phylogenetic distance (separately for each trait), for example as done by Langille et al. (2013, Nature Biotechnology, DOI:10.1038/nbt.2676). I would further advise omitting OTUs from the analysis that have no sufficiently closely related cultured strain (perhaps separately for each trait, depending on the cross-validation results, as per the authors' judgement). As it currently stands, it is hard to give trust to the majority of the authors' trait predictions. Especially for traits where the authors claim an absence of a significant trend, this could be the mere outcome of an extremely inaccurate (essentially random) state prediction.

We agree that our original approach to inferring unknown trait values was a limitation of the original manuscript, and we thank you for motivating us to improve it. We believe our revised approach stands on much firmer footing. As per your suggestion, we now model trait dissimilarity as a function of phylogenetic distance, and drop predictions that fall below a given threshold of phylogenetic distance for each trait (Fig. S3, Table 2). This validation process led us to drop values for four traits (Aggregation score, pH optimum, Salt optimum, and IgA binding affinity) that could not be inferred confidently at any phylogenetic distance. For the remaining traits, the proportion of sequences in our study with data that were either "directly observed" (i.e., drawn from the literature) or inferred within acceptable phylogenetic distances and was always over 75 %.

This validation process (and the associated literature) had a second effect on our manuscript: because 16S gene copy number (GCN) could not be safely predicted for nearly 25 % of the sequences in our study, we no longer correct for 16S GCN in our OTU abundance table. We now concur with Louca et al. (2018) that correcting for 16S GCN is an unresolved problem, and that, with current information validated with phylogenetic data, the best choice is to not make any correction. In addition, we had to drop “biofilm formation” as a trait because it was calculated using BugBase (Ward et al. 2017*), a software tool that automatically corrected for 16S GCN (which we no longer wanted to do), and therefore did not have the statistical rigor of our new approach.

*Ward, T., J. Larson, J. Meulemans, B. Hillmann, J. Lynch, D. Sidiropoulos, J. Spear, G. Caporaso, R. Blekhman, R. Knight, R. Fink, and D. Knights. 2017. BugBase predicts organism-level microbiome phenotypes. *BioRxiv* 1:1–19.

3. Lines 270-272: Related to my previous comments, could it be that the absence of a trend in 16S gene copy numbers over time is due to the high uncertainty in 16S GCN predictions? That is, if 16S GCN prediction is essentially random (as argued by Louca et al. 2018, *BMC Microbiome*), this would mask any real biological signal. Again, a validation of your HSP would add confidence in your trait-based analyses.

After applying our new analytical approach (weighted squared-change parsimony for phylogenetic inference of traits), there was steady decrease in mean 16S GCN over time (Fig. 2), consistent with many other microbial successional systems.

4. Lines 243-246: It is hard to see whether trait-based patterns are really statistically independent solely based on the correlation coefficients without providing any measure of significance. For example, Pearson correlation coefficients between -0.5 and 0.5 can still contain substantial biological/ecological information. How many of the pairwise correlations between -0.5 and 0.5 were statistically significant?

We now provide measures of significance for all trait correlations (see Figure S6). In addition, these values bolster confidence in the correlation coefficients as a result of our improved analytical pipeline.

After performing a Bonferroni correction for multiple comparisons, 48 of the 55 possible pairwise trait pairs were significantly correlated. Therefore, we updated the discussion of our results in two ways: 1) we now consider trait syndromes – i.e., trait values that frequently co-occur in taxa, and may reflect strategies that have been evolutionary successful – and identify a few trait syndromes that may exist in our data (e.g., an inverse relationship between sporulation ability and oxygen tolerance); and 2) we temper our discussion to acknowledge that a given trait pattern may simply be a statistical artifact due to correlation – e.g., the observed decrease in mean GC content over succession may simply be because it correlates (for unclear reasons) with oxygen or temperature tolerance. See lines 292-305 in the new manuscript.

5. Meaning of ecological drift (e.g. lines 353-355 and lines 409-410): The authors should be more cautious when interpreting "randomness" as "ecological drift". Ecological drift usually refers to a very specific type of randomness, namely random fluctuations of population size due to the stochasticity of birth & death events in finite (small) populations. Ecological drift is much more relevant for larger organisms (e.g. plants) which have smaller population sizes, and is arguably much less relevant for typical microbial communities. Many of the seemingly random fluctuations observed by the authors may be due to ecological processes (other than drift) not captured by the handful of traits monitored by the authors, such as phage-host interactions.

Thanks for bringing up this potential point of contention. We are now more precise with our use of ecological drift. As we now state in the manuscript, we agree that gut microbial communities should be less susceptible to drift than other communities due to their relatively larger population sizes. However, the large number of rare taxa in the gut may mean their populations are still vulnerable to stochastic variation. We also now mention that variable immigration rates may interact with drift to drive functionally-neutral changes in community composition. We now include the insightful example of phage-host interactions as a possible explanation for unexplained taxonomic fluctuations. Refer to lines 403-411 in the new manuscript.

6. Comparison of taxonomic and functional convergence (e.g. lines 380-384): The authors find that convergence over time was stronger in trait-based composition than in OTU-based composition, however the authors use different distance metrics for the two (Euclidean vs Bray-Curtis). It's a bit like comparing apples to oranges. The relationship between Euclidean and Bray-Curtis distance is non-linear, and hence it is very hard to compare compositional turnover rates between the two metrics. At the very least I would expect the authors to use the same metric for both OTU- and trait-based distance, for example by treating CWMs like OTU abundances and using weighted Bray-Curtis for CWMs and OTUs. Even better would be to try both metrics and see if the conclusions are robust with regards to the choice of metric. As I mentioned before, I sympathize with the authors' conclusions, but I don't believe they are justified by the present analyses.

Thank you for this comment, and for encouraging us to strengthen the statistical basis of our conclusions. First - because we agreed that it is not initially clear whether Euclidean and Bray-Curtis dissimilarities are comparable – we performed a simulation to better understand how these metrics covary. Specifically:

- 1) We generated an artificial community with 1000 OTUs, a Poisson lognormal abundance distribution, and randomly-assigned trait values, drawn from our trait database, for each of the traits in our analysis.**
- 2) We generated 4394 mock communities (equal in size to the original artificial community) with the full range of possible Bray-Curtis dissimilarities. I.e., communities ranging from having all individuals of OTUs present in the original artificial community (Bray-Curtis**

- dissimilarity = 0), to having no individuals of OTUs in the original artificial community (Bray-Curtis dissimilarity = 1).
- 3) We then calculated Bray-Curtis dissimilarity, OTU-based Euclidean dissimilarity (as per the suggestion of Reviewer 1), and trait-based Euclidean dissimilarity between each mock community and the original artificial community.
 - 4) The relationships between Bray-Curtis dissimilarity and both Euclidean metrics were in fact linear, supporting the conclusion that trait-based convergence occurred faster than OTU-based community convergence. In the plot below, dissimilarity metric data are shown as black (overlapping) circles, and fitted linear models are shown as red lines.

While this was just an informal test of the relationship between Bray-Curtis dissimilarity and Euclidean distance, we feel this simulation-based exploration provides confidence in our conclusions. We welcome additional information about the relationship between these metrics.

Nonetheless: because the comparison of trait-based convergence and OTU-based convergence is one of the main features of the manuscript, we also identified an independent, statistically rigorous approach to address our question. Briefly, we compared observed rates of trait-based convergence to null model predictions; null model predictions were generated by performing the same calculations, but with a version of the trait data for which trait values were randomly permuted among OTUs. This approach firmly supported that trait-based rates of convergence exceeded those expected by chance, given the observed rates of OTU-based convergence. See the updated version of Figure 4 and Lines 253-261 in the revised manuscript for further details.

7. Figure 2 - 16S gene copies: It looks like there may be a trend towards fewer 16S gene copies over time in the first few weeks, which would be consistent with standard theory. In other words, high-16S-copy strains (r-strategist) have an advantage at first, but are quickly replaced by K-strategists within just a few weeks. Hence, could it be that the time scales at which some of these traits vary are much smaller (or much longer) than the sampling resolution (or sampling duration, respectively)? I think this caveat needs to be discussed in the article.

After revising our approach (weighted squared-change parsimony for phylogenetic inference of traits, see above), there is now a decrease in mean 16S GCN over succession (Fig. 2).

Minor comments (i.e. can easily be addressed)

How does your work relate to that recently published by Ferretti et al? *

Perhaps a brief discussion is worthwhile.

* Ferretti et al. (2018). Mother-to-Infant Microbial Transmission from Different Body Sites Shapes the Developing Infant Gut Microbiome. *Cell Host & Microbe* 24:p133–145.e5

Indeed! We now mention this interesting study in our discussion. Refer to lines 354- 357 in the revised manuscript.

Please be more clear when referring to the Greengenes phylogeny (or taxonomy?). Did you use a taxonomic tree or a 16S-based phylogenetic tree (the two may not be congruent). For example, if you used a phylogenetic tree for generating Fig. S2, how did you treat nodes/subtrees that were non-monophyletic taxonomically speaking?

We now use a 16S-based phylogenetic tree, as stated in the revised manuscript. Refer to lines 176 – 178 in the revised manuscript.

Fig. S3: Please include information on the statistical significance of those correlations (either in the same figure, e.g. as embedded numbers) or in another corresponding figure.

Done.

Please provide a summary table as supplemental material, listing state predictions for each Greengenes OTU as used in this study. This will be useful for future studies that may want to reproduce your results or extend your approach to other cases. In addition, since your state predictions may contain substantial error, it is important for future investigations to be able to disentangle what's a true biological result vs an artifact of your particular HSP.

State predictions are now in supplementary file "trait_value_predictions.csv". The inclusion of this file is referred to on line 151 in the revised manuscript.

Line 161: Figure number S3 seems wrong here, as Fig. S3 shows correlation coefficients, not coverages.

Fixed.

Lines 256-269: I'm not sure I follow your reasoning about why sporulating taxa increase with time. If sporulation presents a mechanism for dealing with oxidative stress, why would this be more important at a latter successional stage, when the gut environment becomes less aerobic overall?

Our argument was that sporulation improves the probability of successful dispersal among hosts, leading to an accumulation (and proliferation) of sporulating taxa as infants age. We have added an additional supplementary figure (Figure S7) to illustrate this point, and hopefully better articulated this argument. Please refer to lines 315-235 in the revised manuscript.

Lines 291-293: This sentence was very confusing to me. Could you improve the wording?

Done.

Line 309: Please clarify at first mention what "variance" refers to (e.g. variance across individuals of the same age).

Done.

Line 348: "Such an explanation": Please clarify what "such" refers to (daily variations in diet, or cyclical community dynamics?).

This sentence has been rephrased.

Lines 423-424 ("illustrating that gut communities are not infinitely functionally resistant"): I would add "and/or that tradeoffs exist between antibiotic resistance and other traits".

Agreed. The phrase has been added.

Axis of Figure 1: I recommend clarifying in the left-most Y-axis label that OTUs are ranked by *time of* their peak global abundance (this only became clear to me after reading the figure caption).

The panel in question from Fig. 1 was removed, because, after revisions, we felt it did not directly add to our research questions.

Reviewer #2 (Remarks to the Author):

In this paper entitled TRAIT-BASED COMMUNITY ASSEMBLY AND SUCCESSION IN THE INFANT GUT MICROBIOME, Guittar et al apply a trait based approach to look at how microbial traits transmission over the course of succession in the human infant gut. The study leverages data produced from two published studies connected to the DIABIMMUNE study group focused on the effects of antibiotics (Yassour et al. 2016) and type-1 diabetes (Kostic et al. 2016) on gut community succession. Overall, this paper is really very interesting. It is very well written. It has a strong conceptual basis and applying a trait based approach to understand infant microbiome succession is novel. I think that the conclusions made from the data analyses are generally insightful and well presented. Additionally, the authors do well in including discussion of findings from unrelated studies and how this work provides explanations or support to the conclusions of those other studies. I thought, however, that the paper had weaknesses in methodological description and statistical rigor as detailed below.

Thank you for your time in reviewing the work and your positive words, and for highlighting areas needing improvement.

Major Revisions

1) The description of methods that were used to generate and process the microbiome data used in this study has some major gaps. Most notably lines 105-107 where it says “Briefly, stool samples of infants were collected by participants’ parents, processed at the DIABIMMUNE Core Laboratory, and sequenced at the Broad Institute for DNA extraction on the Illumina HiSeq 2500 platform, with ~2.5 Gb of sequence per sample and 101 bp paired-end reads.” Is highly inadequate and really needs to be greatly expanded. It is not even clearly stated whether this is 16S rRNA amplicon or shotgun metagenomic data. Details on DNA extraction methodology, sample collection and storage, how the sequencing was conducted, how the sequence data was bioinformatically processed to e.g. remove noise, call OTUs, assign OTUs taxonomically are all completely absent.

Thank you for this comment. We significantly expanded the description of methods, and, given that this work presents a re-analysis of existing published data, also now clearly point reader to the original published datasets and their descriptors.

a. Related to this comment on line 151, it is unclear how these “Latin binomial” assignments are being made.

Thank you for pointing this out. At the great suggestion of Reviewer 1, we have greatly refined our process of assigning trait values to OTUs using a weighted squared-change parsimony approach for phylogenetic inference of traits. Specifically, we no longer use the Greengenes phylogeny (we agree it is now outdated). Instead, we re-processed the raw 16S rRNA gene sequencing data using a USEARCH pipeline, and mapped the resulting OTUs to the 123 release of SILVA. We then mapped trait values to OTUs (only) when both Genus and Species labels matched exactly. We have updated the methods to clarify this point. We have also provided our computing workflow on GitHub so that readers can reproduce our methods exactly (https://github.com/ShadeLab/microbiome_trait_succession).

Related to this, we now also link literature-derived trait values to taxa from the Living Tree Project (release 132), a well-curated, up-to-date set of 13900 16S rRNA SSU sequences from type specimens with confirmed Genus and Species taxonomic identities. Thanks to the well-resolved taxonomic information of taxa from the Living Tree Project, we were able to map many more trait data onto its (16S-derived) phylogeny, dramatically improving our ability to infer unknown trait values through hidden state construction. We have updated our methods to reflect this change as well.

2) This paper is pooling data from 2 different studies – with a focus on antibiotics and Type I diabetes. Did these studies both use the same methods throughout? What were the criteria for recruiting for these different studies – e.g. did they focus on infants with particular risk attributes? To what degree might these criteria affect the degree to which the findings here are generalizable?

Thank you for this comment, which made us realize that we could clarify our description of how the studies were aggregated. We have added information about the infant cohorts from each of the two studies to our methods, and they have generally used the same methods throughout because they were collected as part of the same larger research project . We also explained why we pooled infants with variable characteristics and/or histories that are known to affect gut community composition. Because our foremost aim was to characterize general patterns of gut primary succession, we are looking for patterns that hold true despite host-related differences.

3) Figure S2 shows phylogenetic distances used when inferring traits. To what extent are these traits conserved at these distances? In some cases like temperature optimum, 50% of the inferences are

being made at the class level. It seems to me that there should be some sort of cut off where no inference is made because the inference is uninformative at that phylogenetic distance.

Agreed. We now drop hidden state predictions when inference was limited. Please see our response to reviewer 1.

4) In general, this paper lacks much statistical analysis of reported trends which I think is a major weakness. E.g. Figure S3 could indicate which stars which correlations are statistically significant. Figure S4 could indicate which traits significantly differed across early, mid and late succession. There are some other areas noted below where I think statistics would better support their arguments.

Thank you for this comment. We have improved the statistical rigor and support of our results in several areas, including the suggested changes to Figure S3 and S4.

5) The cohort's characteristics are not well described which sometimes makes their results difficult to understand. As an example, on lines 371—372, the authors make a point about infants shifting from diets subsisting only on milk to solid foods, but it is not clear to me based on their cohort description the degree to which these infants are fed breast milk versus formula.

Thank you for this comment, and we realize now that we needed to improve clarity in this description. We have rephrased this sentence to more precisely include the nuance that infants could have been weaned from more easily digested breast milk or formula to more difficult to digest solid foods. The distinction between breast milk and formula-fed infants is not apparent from our analysis.

6) It is unclear to me how the authors calculated the “mean trait value” mentioned in line 176 from the methods as written.

Thank you, we have clarified this sentence.

Minor Revisions

1) With regards to the point about a lack of a trend in 16S rRNA copy number: (lines 275-278), there does seem to be a trend that this is highest in the first 2 months of life, which could indicate that it

is important in very early succession. I do feel like having some sort of statistics to make this point with more confidence would be nice though.

After refining our approach at the great suggestion of Reviewer 1 (using weighted squared-change parsimony for phylogenetic inference of traits, and dropping predictions that were no better than random guesses), there is now a visible decrease in mean 16S GCN over succession (Fig. 2). We focused our results on reporting the overarching trends and confidence intervals so that the major shifts for each trait could be clearly identified from these trends. Please also refer to prior comment to reviewer 2, above.

2) With regards to the analysis of cesarian section versus vaginal delivery (lines 282-291), what was striking to me about these data in Figure 4 is just how much more variable the traits are in C-section versus vaginal delivery. Again, might be nice to have a statistical test that indicates that variance in traits is significantly greater, but based on the depicted confidence intervals this should be the case...

Thank you for this insightful comment. Notably, the cohort of C-section infants was smaller than the cohort of vaginally delivered infants. We have performed statistical tests on these data (see the Fig. S9 legend), and they are indeed significant despite that the C-section infants have more variable trait values, as evidenced by the right panel in Figure S9.

3) When defining the discrete time intervals (early vs. mid vs. late) in lines 164-170 the correlations with time appear to be reversed. For example, Figure 1 shows increasing abundances over time for the late succession OTUs which would appear to be positively correlated with time.

Thank you for catching that! You are very right. This gaffe has been corrected.

4) Typos/grammar

Thank you for catching these. We have carefully edited throughout for typos and grammar errors.

a. Line 112: should be "as a potential confounding variable"

Fixed. (We ended up leaving in an infant from Estonia which we had originally removed, and changed the sentence accordingly.)

b. Line 150: “similar” should be “similarity”

This sentence has been removed when updating our methods section.

c. Line 178: the abbreviation (GAMs) should be inserted here

We have removed this sentence.

d. Line 237: “community weighted means” should just be “CWM”

Fixed.

e. Line 323: “(3)” is needed before “trait based Euclidean...”

Fixed.

5) Line 161: Figure S3 is referenced but it should be Figure S2.

Fixed.

Reviewers' Comments:

Reviewer #1:

Remarks to the Author:

The authors have addressed all of my concerns. I now find the manuscript much stronger, and I recommend publication. Congratulations on this nice study!

Reviewer #2:

Remarks to the Author:

Overall I think the paper is greatly improved and is really nice work. The authors addressed all of my concerns and I also like the changes that they made in response to comments of Reviewer 1. I just note below a couple of minor edits:

- 1) lines 80-81. The words "and type-1 diabetes" seem a little out of place in this sentence.
- 2) Line 112. There is a missing closed parentheses
- 3) Line 114: "development" is misspelled
- 4) Line 165: ranging from zero to what?
- 5) Line 337: Should "@" be there in reference?

Thank you for your positive reviews and agreeing to publish a revised version of our original manuscript in Nature Communications! Our responses to the remaining reviewer concerns are below and in bold.

Reviewers comments and our responses.

Reviewer #1 (Remarks to the Author):

The authors have addressed all of my concerns. I now find the manuscript much stronger, and I recommend publication. Congratulations on this nice study!

Thank you to Reviewer 1 for the positive review of our revised manuscript!

Reviewer #2 (Remarks to the Author):

Overall I think the paper is greatly improved and is really nice work. The authors addressed all of my concerns and I also like the changes that they made in response to comments of Reviewer 1. I just note below a couple of minor edits:

- 1) lines 80-81. The words “and type-1 diabetes” seem a little out of place in this sentence.
- 2) Line 112. There is a missing closed parentheses
- 3) Line 114: “development” is misspelled
- 4) Line 165: ranging from zero to what?
- 5) Line 337: Should “@” be there in reference?

Catherine Lozupone

These typos/mistakes have been fixed, and we have also read over the entire manuscript several additional times and fixed a few additional typos and unclear sentences. Many thanks to Reviewer 2 (Dr. Catherine Lozupone) for the positive review of our revised manuscript!